

# Colonization of novel algal habitats by juveniles of a marine tube-dwelling amphipod

Marilia Bueno[1], Glauco B.O. Machado[1,2] and Fosca P.P. Leite[1]

[1] Departamento de Biologia Animal, Universidade Estadual de Campinas, Campinas, São Paulo, Brazil
[2] Instituto de Biociências, Campus do Litoral Paulista, Universidade Estadual Paulista, São Vicente, São Paulo, Brazil

## ABSTRACT

**Background:** Dispersal is an important process affecting population dynamics and connectivity. For marine direct developers, both adults and juveniles may disperse. Although the distribution of juveniles can be initially constrained by their mothers' choice, they may be able to leave the parental habitat and colonize other habitats. We investigated the effect of habitat quality, patch size and presence of conspecific adults on the colonization of novel habitats by juveniles of the tube-dwelling amphipod *Cymadusa filosa* associated with the macroalgal host *Sargassum filipendula*.

**Methods:** We tested the factors listed above on the colonization of juveniles by manipulating natural and artificial plants in both the field and laboratory.

**Results:** In the laboratory, juveniles selected high-quality habitats (i.e., natural alga), where both food and shelter are provided, when low-quality resources (i.e., artificial alga) were also available. In contrast, habitat quality and algal patch size did not affect the colonization by juveniles in the field. Finally, the presence of conspecific adults did not affect the colonization of juveniles under laboratory condition but had a weak effect in the field experiment. Our results suggest that *C. filosa* juveniles can select and colonize novel habitats, and that such process can be partially affected by habitat quality, but not by patch size. Also, the presence of conspecifics may affect the colonization by juveniles. Successful colonization by this specific developmental stage under different scenarios indicates that juveniles may act as a dispersal agent in this species.

Corresponding author
Marilia Bueno,
mariliabueno@live.com

## INTRODUCTION

Dispersal of individuals has important consequences for population dynamics, connectivity (*Hansson, 1991*; *Ronce, 2007*; *White et al., 2019*) and ecosystem function in aquatic environments (*Little, Fronhofer & Altermatt, 2019*). This process has been related to ultimate and proximate causes (*Bowler & Benton, 2005*; *Burgess et al., 2016*), and also depends on organismal traits that often vary among individuals within a population, such as body size, sex, and developmental stage (*Lawrence, 1987*; *Munguia, Mackie & Levitan, 2007*;

*Munguia, 2015*). A variety of strategies related to dispersal are found among aquatic invertebrates, involving passive and/or active components and allowing the movement of these organisms across short and long distances (*Martel & Chia, 1991*; *Palmer, Allan & Butman, 1996*; *Kinlan & Gaines, 2003*; *Ptatscheck et al., 2020*).

For many sessile and mobile benthic species, planktonic larvae represent the main dispersal stage, often exploiting resources in a habitat different from where adults are established (*Mileikovsky, 1972*; *Grantham, Eckert & Shanks, 2003*; *Gaines et al., 2007*; *Weersing & Toonen, 2009*). However, dispersal during other life stages may have important implications for population dynamics (*D'Aloia et al., 2017*), with juveniles and/or adults playing important roles on the dispersal process. For mobile benthic species with direct development (i.e., lacking a larval stage), such as amphipods and isopods, transport movements occur over both short and long distances during the whole lifespan (*Waage-Nielsen, Christie & Rinde, 2003*; *Kumagai, 2006*; *Davidson, Rumrill & Shanks, 2008*). In this case, the spatial distribution of newborn juveniles can be initially constrained by the female's habitat choice (*Poore & Steinberg, 1999*), with individuals of different developmental stages often sharing the same habitat (*Thiel, 1999*; *Brooks & Bell, 2001*; *Poore, 2004*; *Miranda & Thiel, 2008*).

The potential dispersal of direct developers may vary with age, with drastic consequences for population dynamics (*Munguia, Mackie & Levitan, 2007*; *Bringloe et al., 2013*). For some species, adults have an important role on the dispersal process, while juveniles can be the main dispersal stage for other direct developers (*Franz & Mohamed, 1989*; *Bueno & Leite, 2019*). Even when both developmental stages can disperse, adults and juveniles may be under different pressures to emigrate and colonize new patches (*DeWitt, 1987*; *Poore, 2004*). For instance, adults may be motivated to disperse in order to find mates to copulate (*Bringloe et al., 2013*) whereas distribution and dispersal of juveniles may be strongly affected by intraspecific competitive interactions. The presence of conspecifics in a patch may be indicative of habitat quality, attracting juveniles to settle (*Bringloe et al., 2013*; *Drolet et al., 2013*); alternatively, the high-density of conspecific adults can inhibit the settlement of juveniles (*Wilson, 1989*; *Jensen & Kristensen, 1990*). For some tube-dwelling amphipods, large individuals can also be aggressive towards smaller ones (*Connell, 1963*; *Brawley & Adey, 1981*), probably forcing them to search for new sites (*DeWitt, 1987*).

Yet, juveniles and adults of brooding invertebrates may differ in their ability to exploit habitats with different qualities, resulting in distinct patterns of colonization among these developmental stages. For example, juveniles of the herbivorous amphipod *Sunamphitoe parmerong* (formerly *Peramphithoe parmerong*) mainly inhabit the high-quality algal food *Sargassum linearifolium*, while adults are able to occupy that host and also to colonize the poor-quality algal food *Padina crassa* (*Poore, 2004*). Juveniles of the amphipod *Pontogammarus robustoides* consistently prefer inhabiting natural macrophytes over artificial ones, while adults do not distinguish among these habitat choices (*Czarnecka, Kobak & Wiśniewski, 2010*). These examples suggest that the colonization of juveniles may be affected by the habitat quality. Furthermore, large patches can attract more adult amphipods than small ones, while the colonization of juveniles may

be unaffected by the patch size, probably because of the small size of these individuals (*Bueno & Leite, 2019*). In this sense, understanding the role of dispersal on the population dynamics and connectivity of direct developers requires investigating the factors driving the colonization of specific developmental stages.

Marine macroalgae harbor a diverse fauna in shallow waters (*Christie, Norderhaug & Fredriksen, 2009*; *Gan, Tay & Huang, 2019*). Epifaunal assemblages are maintained through the developmental cycle of their hosts, which provide continuous renewal of the resource (i.e., food and/or shelter), allowing associated populations to thrive (*Seed & O'Connor, 1981*). Among mobile epifauna, amphipods are highly abundant on macroalgal beds and can show colonization within few days on defaunated algal thalli (*Taylor & Cole, 1994*; *Tanaka & Leite, 2004*). We selected that system to investigate the factors driving the colonization of novel habitats by juveniles. For that, we investigated the effect of habitat quality, algal patch size and presence of conspecific adults on the colonization of juveniles of the tube-dwelling amphipod *Cymadusa filosa* Savigny, 1816 (Amphipoda: Ampithoidae). We used this developmental stage because a previous investigation reported the important role of juveniles on the dispersal of amphipod species in the study system (*Bueno & Leite, 2019*). First, by manipulating natural and artificial plants in both laboratory and field, we tested if juveniles are able to select and colonize a new algal patch depending on the habitat quality. Because juveniles seem to be sensitive to the habitat quality (*Poore, 2004*; *Czarnecka, Kobak & Wiśniewski, 2010*), we hypothesized that they would colonize high-quality habitats more frequently than low-quality habitats. Second, we tested if the algal patch size affects the colonization rate by juveniles in the field. Because of their small body size, we expected that juveniles would be indifferent to the algal patch size. Finally, under laboratory and field conditions, we tested if the presence of conspecific adults on algal patches affects the settlement of juveniles. We expected that colonization by juveniles would be negatively affected by the presence of adults as a result of intraspecific aggressive interactions (*Connell, 1963*; *Brawley & Adey, 1981*).

## MATERIALS AND METHODS

### Organisms and study area

The tube-dwelling amphipod *Cymadusa filosa* is a mesograzer commonly found in association with a variety of macroalgae in marine shallow waters (*Tanaka & Leite, 2003*; *Appadoo & Myers, 2004*; *Bueno, Dias & Leite, 2017*; *Machado, Ferreira & Leite, 2019*). As a generalist herbivore, *C. filosa* is able to feed and survive on different algal diets (*Machado, Siqueira & Leite, 2017*, *Machado, Ferreira & Leite, 2019*). This mesograzer often builds tubes on algal blades, with newborn juveniles settling in the vicinity of their mother's tube (*Appadoo & Myers, 2003*). Organisms used in the field and laboratory experiments were collected on a rocky shore (depth of 1–2 m) at Fortaleza Beach (23°32′S, 45°10′W), Ubatuba, São Paulo State, Brazil (SISBIO/ICMBio # 51999-1). The brown alga *Sargassum filipendula* is abundant on the subtidal level of this rocky shore and shelters a highly diverse amphipod assemblage, including *C. filosa* (*Jacobucci, Tanaka & Leite, 2009*).

# EXPERIMENTS: GENERAL PROCEDURES

Field experiments were conducted during March 2017 (late summer). Natural *Sargassum* thalli used in the experiments were collected at the study area and cleaned of epibionts in the laboratory. For all field experiments, each experimental unit (EU) included one 70 cm iron stake (6 mm diameter) driven into the sediment to serve as a foundation for the whole unit. A squared plastic screen (8 cm × 8 cm) with 1 cm mesh size was attached to the iron stake with a long zip tie; it supported the algal patch, which was artificial or natural, according to the experiment. A small float was attached to the iron stick and the squared plastic screen by a rope to hold the whole unit erect (see Fig. 1 and *Bueno & Leite, 2019* for further details). Algal substrates were attached to the plastic screen with nylon wires. We randomly deployed EUs parallel to shore ~ 1 m from the shoreline, leaving at least 2 m spacing between adjacent EUs. After 48 h, experimental algal thalli were placed in bags with mesh size of 200 μm, transported to the laboratory and frozen. In the laboratory, samples were washed under freshwater to separate amphipods, which were preserved in ethanol 70%, counted under a stereomicroscope and identified to species level according to the literature (*Ruffo, 1982*; *Barnard & Karaman, 1991*; *LeCroy, 2000*, *2002*, *2004*, *2007*, *2011*).

All laboratory experiments were performed in a climate-controlled room at 22 °C using a 12:12 h photoperiod. *Cymadusa filosa* individuals were obtained from macroalgal hosts from the study area and, eventually, from other rocky shores nearby. After collection, epiphytes and associated fauna were carefully removed from algal fronds in the laboratory. *Cymadusa* individuals were kept in a stock culture (20 L plastic tank) with seawater and continuous aeration. We fed amphipods with fresh *Sargassum* every other day and changed the water once a week. For each experiment, juveniles were obtained from ovigerous females collected in the stock culture. Each replicate (a 20 L plastic tank with aerated seawater and algal thallus) was conducted with juveniles obtained from the same female. Ideally, the experiments should be conducted with females carrying juveniles in their marsupium, which would be released on the experimental habitats. However, incubation period is variable among females and juveniles can leave the marsupium at different times (*Sheader & Chia, 1970*). In this case, we attempted to minimize the difference in the size and age of juveniles within and among replicates by conducting the trials with the smallest individuals on the third day after the female had molted. For that, each ovigerous female was kept in a beaker with seawater (150 ml) until juveniles were released from the female's brood pouch (i.e., after the ovigerous female had molted). Then, newborn juveniles were assigned to a replicate.

## Habitat quality

Field and laboratory experiments were conducted to test the effect of habitat quality on the colonization of *C. filosa* juveniles. For that, we used natural *Sargassum* thalli as high-quality habitat, because it offers both food and shelter for *C. filosa* (*Machado, Ferreira & Leite, 2019*), and artificial *Sargassum* thalli as low-quality habitat, because it could provide only a shelter for these herbivorous juveniles. Artificial *Sargassum* fronds (Bio Models Co.) were physically identical to *Sargassum* thalli in shape, size, branching and

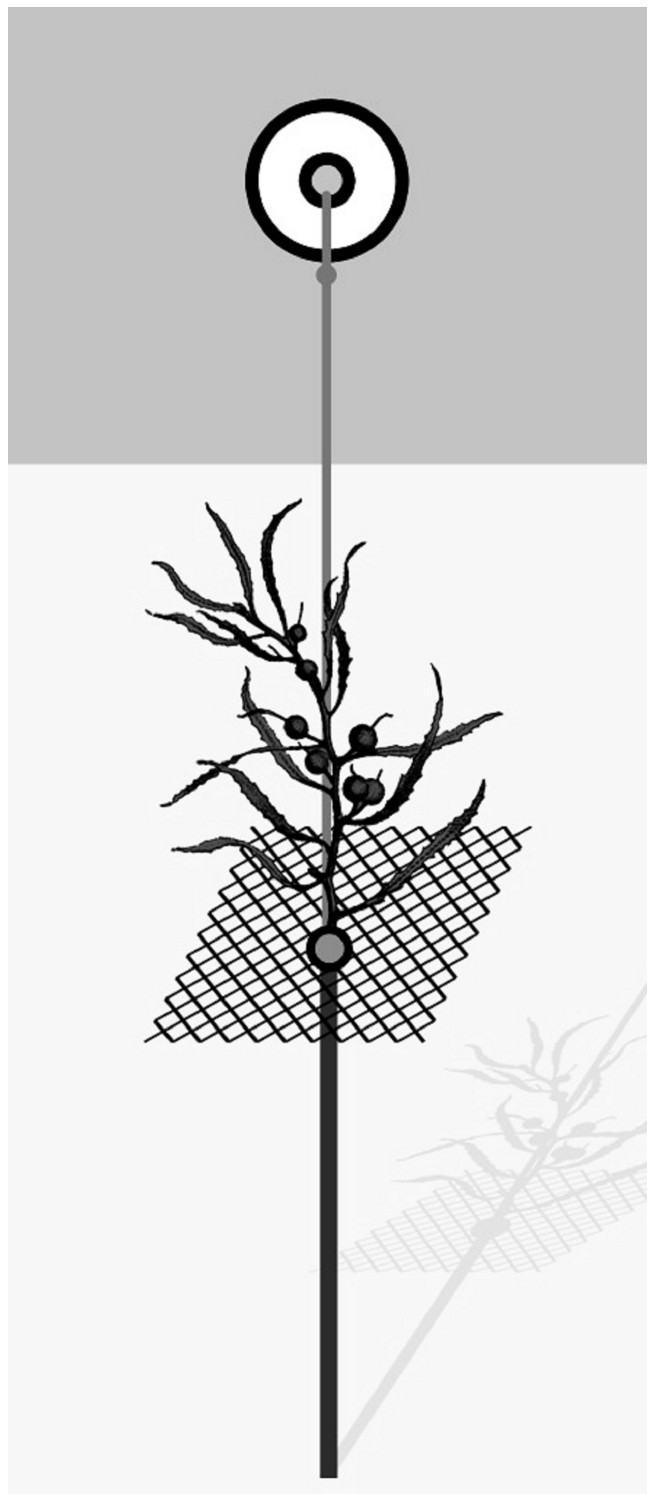

**Figure 1 General scheme of the experimental unit used in field experiments.** From *Bueno & Leite (2019)*.

color and were arranged to have similar surface area to the natural algae. We estimated algal surface area by measuring the area of flattened branches with transparent adhesive tape in a PVC plate. We used the software Image-J to calculate area and duplicated the values to account for the front and back of branches. In the field, artificial ($108.2 \pm 3.0$ cm$^2$) and natural ($115.8 \pm 25.8$ cm$^2$) algae were attached to the squared plastic screen of the experimental structures ($N = 5$). No difference in area was detected among artificial and natural algae (GLM, Gamma distribution, $P = 0.518$). The field experiment was performed as described above in general procedures. The number of *C. filosa* juveniles on each experimental algal thallus was used as the response variable.

In the laboratory, besides manipulating the quality of the novel habitat, we also manipulated the quality of the source habitat (i.e., the habitat where juveniles were before the novel habitat was offered). For that, we used four levels of combination of source/novel habitats: natural/natural, natural/artificial, artificial/natural and artificial/artificial ($N = 6$). For each replicate, we placed the first algal thallus (i.e., source habitat) in a 20 L plastic tank with aerated seawater and added the juveniles using a pipette. After 30 min of acclimatization, we added the second algal thallus (i.e., novel habitat). Both algal thalli were kept 5 cm apart and hold erect by using a fishing buoy at the top and a fishing rod at the bottom, both tied with nylon wires. We used similar number of juveniles in each replicate ($18 \pm 2$ juveniles) to avoid confounding density-related factors. We counted the juveniles on each algal thallus after 24 h by carefully removing the algal fronds from the tank. We then calculated the proportion of juveniles that colonized the novel habitat (i.e., response variable). After each experimental run, we filtered all the water from the tanks and cleaned the algal fronds to ensure that there were no remaining juveniles.

## Algal patch size

A field experiment was performed to test the effect of algal patch size on the colonization of *C. filosa* juveniles. For that, the experiment consisted of two treatments representing algae with small ($30.8 \pm 13.8$ cm$^2$) and large ($115.8 \pm 25.8$ cm$^2$) surface area ($N = 5$). *Sargassum* fronds with small and large surface area without any epibionts were attached to the square plastic screen on the experimental structures in the field, then the experiment was conducted as described above. The number of *C. filosa* juveniles on each experimental algal thallus was used as response variable. We estimated algal surface area applying the same methodology described above.

## Presence of conspecific adults

To test the effect of conspecific adults on colonization by *C. filosa* juveniles, we performed field and laboratory experiments manipulating the abundance of adults on algal thalli. In both experiments, we used only male adults to avoid mating between males and females. For the field experiment, to test how the presence of *C. filosa* adults affects juveniles' colonization, we kept the algal thallus with adults inside of a 250 ml acrylic cup. We perforated the walls and the lid of the acrylic cups using a drill with a fine bit, producing 1 mm diameter orifices. The orifices allowed water and juveniles to pass (and perhaps small adults). In a laboratory pilot trial, we confirmed that *C. filosa* juveniles

were able to pass through the wall of the cup through the orifices. To allow adult males to build tubes and settle in the *Sargassum* fronds, we kept them in cups with seawater and algal patches for 24 h in the laboratory before beginning the field experiment. Each experimental set was composed of one algal patch (~3.6 g of wet mass) with 0, 4 or 8 adult males within an acrylic cup ($N = 4$ or $N = 5$). We attached each experimental set to the squared plastic screens of the experimental structures in the field. The experiment was performed using the same methods described above. After the collection, experimental sets were transported to the laboratory and, before freezing these samples, cups were inspected for the presence of live *C. filosa* adults to assess if the number of active adult males inside the cups had changed throughout the experiment. After 48 h, for the treatments with 0, 4 and 8 male adults added initially, we found an average of 0, 3.6 (±0.5) and 7.5 (±0.6) adults inside the cups, respectively. As a response variable, we used the number of *C. filosa* juveniles in each experimental set.

In the laboratory, we tested whether the effect of adults on the colonization of natural algal patches by juveniles depends on the type of source habitat occupied by juveniles. For that, we manipulated the presence of adults on natural algal thallus (two levels: 0 or 4 adults) and the type of source habitat of juveniles (three levels: no algal thallus, artificial algal thallus and natural algal thallus) in an orthogonal design (N=6 for each level of combination). Considering in the field experiment there was no difference for both treatments with adults regarding their effects on juveniles' colonization (see Results), we decided to use only two levels for adult presence (i.e., 0 and 4 adults). Each replicate consisted of one plastic tank with a natural *Sargassum* patch with or without adult males. *Cymadusa* adult males (15.7 ± 1.6 mm of body size) were obtained from the stock culture, carefully placed on the algal frond using a delicate paintbrush and kept in the plastic tank for at least 6 h for acclimatization. Again, we used a similar number of juveniles in each replicate (16 ± 2 juveniles) to avoid confounding density-related factors. Juveniles were added to perforated acrylic cups similar to those used in the field experiment, placed on one of the types of source habitat. The cup with the juveniles was 5 cm away from the natural algal thallus with or without adults. After 24 h, we counted the number of juveniles and then calculated the proportion of juveniles that colonized the algal thalli outside the cups.

## Data analyses

We used general linear models (GLM) to analyze data from field and laboratory experiments. To compare the abundance of *C. filosa* juveniles among treatments of the field experiments, we used GLM with Poisson ('habitat quality' experiment), Quasi-poisson ("algal patch size" experiment) or Negative binomial ("presence of adults" experiment) distributions. For the field experiment testing the effect of presence of adults, algal wet mass (log-transformed) was used as an offset variable. For the laboratory experiments, we compared the proportion of juveniles that colonized a novel habitat among treatments using GLM with Quasi-binomial distribution. When the effect of a factor was detected, we used Tukey's test to explore possible differences among treatments.

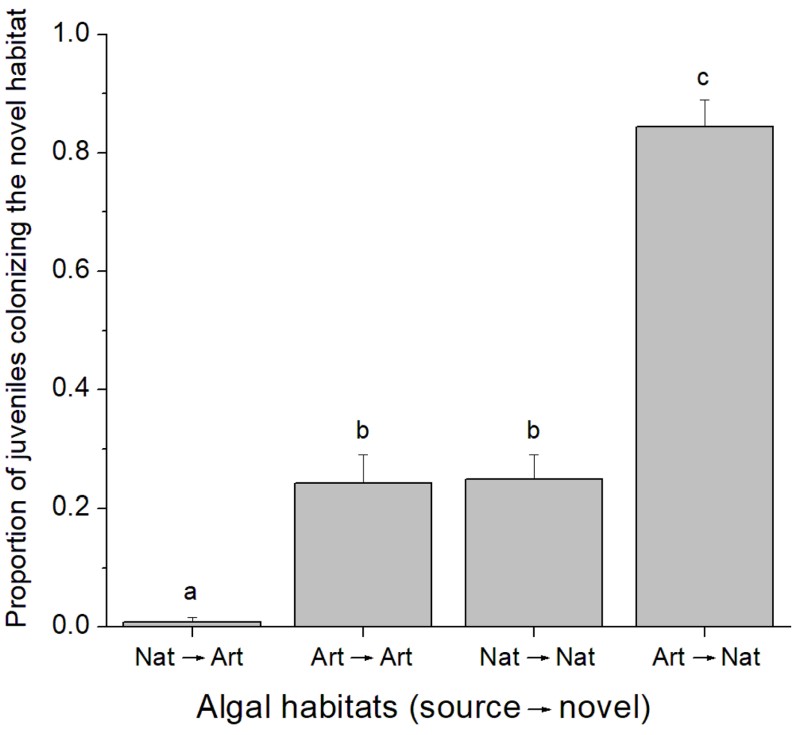

**Figure 2 Proportion of *Cymadusa filosa* juveniles colonizing the novel habitat offered in the laboratory.** Source and novel habitats are indicated as "nat" when natural alga and as "art" when artificial alga. Arrows indicate the colonization path. Bars represent standard error. Different letters indicate significant difference among combinations of source and novel habitats (Tukey test, $p < 0.05$).

## RESULTS

### Habitat quality

A total of 37 juvenile amphipods colonized experimental algal thalli in the field. From this, 17 juveniles were identified as *C. filosa*. Although more individuals were found in natural algal patches, there was no difference in the abundance of *C. filosa* juveniles between artificial ($1.2 \pm 0.8$ juveniles) and natural ($2.2 \pm 1.3$ juveniles) algal thalli (GLM, Poisson distribution, $P = 0.222$). In contrast, in the laboratory experiment, the quality of source and novel habitats (i.e., artificial or natural alga) influenced the colonization rate by *C. filosa* juveniles (GLM, Quasi-binomial distribution, $P < 0.001$). The highest proportion of juveniles colonizing a novel habitat was found when the source was artificial, and the novel habitat was natural alga. In turn, the lowest proportion of juveniles colonizing a novel habitat occurred when the source and novel habitats were a natural and an artificial alga, respectively. Intermediate values of proportion of juveniles colonizing a novel habitat were found when novel and source habitats were of the same quality (Fig. 2).

### Algal patch size

Of 47 juvenile amphipods that colonized the experimental algal thalli, 12 were *C. filosa* juveniles. The abundance of *C. filosa* juveniles was similar between algal patches with small

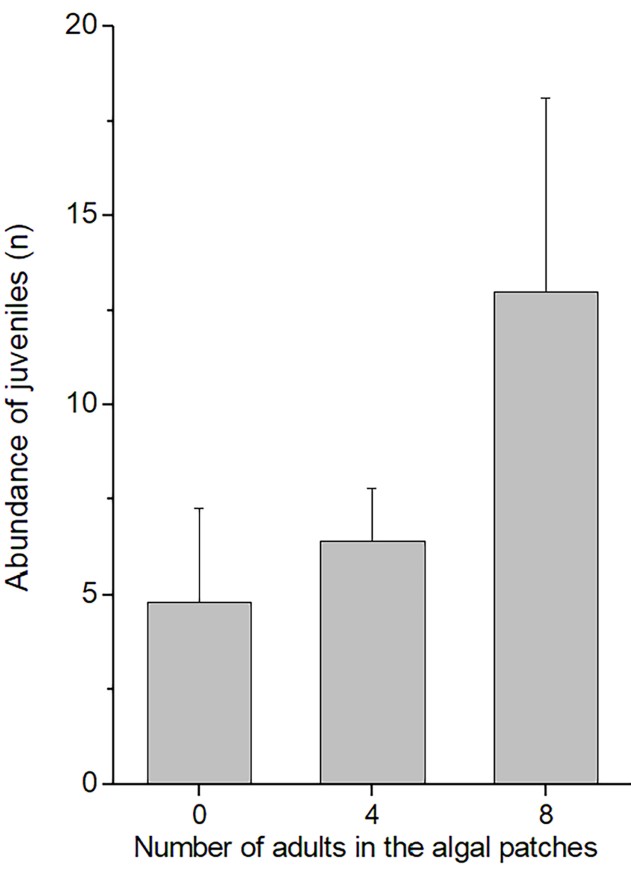

**Figure 3 Abundance of juveniles of *Cymadusa filosa* that colonized algal experimental patches in the field. Different numbers of conspecific adult males (0, 4 and 8) were established in the algal patches when they were deployed in the field.** Bars represent standard error.

(1.2 ± 1.3 juveniles) and large (2.2 ± 1.3 juveniles) surface area (GLM, Quasi-poisson distribution, $P = 0.277$).

## Presence of adults

In the field experiment, 111 *C. filosa* juveniles (of 313 juvenile amphipods in total) colonized experimental algal thalli. Overall, the abundance of *C. filosa* juveniles had a tendency to increase with the density of adults (Fig. 3). However, this effect was only marginally significant (GLM, Negative binomial distribution, $P = 0.055$). For the laboratory experiment, we found a significant interaction between the factors 'presence of adults' and 'type of source habitat' (Table 1). Despite this, from the Tukey's test, we found that differences (or similarities) among levels of one factor did not depend on the levels of the other factor for the meaningful comparisons to this study. *Cymadusa filosa* juveniles without alga or with artificial alga colonized a novel habitat (with or without adults) in a higher proportion than juveniles with natural alga as source habitat (Fig. 4). Within each level of source habitat, the presence of adults did not affect the colonization by juveniles (Fig. 4).

**Table 1 Analysis of Deviance for GLM with Quasi-binomial distribution fitted to the proportion of juveniles from different source habitats colonizing natural algal thallus with or without adults at the laboratory experiment.**

| Source of variation | df | Deviance | Residual df | Residual deviance | F | p |
|---|---|---|---|---|---|---|
| NULL | – | – | 35 | 20.423 | – | – |
| Source habitat (SH) | 2 | 12.181 | 33 | 8.242 | 33.008 | <0.001 |
| Presence of adults (PA) | 1 | 0.779 | 32 | 7.462 | 4.223 | 0.049 |
| SH X PA | 2 | 1.699 | 30 | 5.763 | 4.605 | 0.018 |

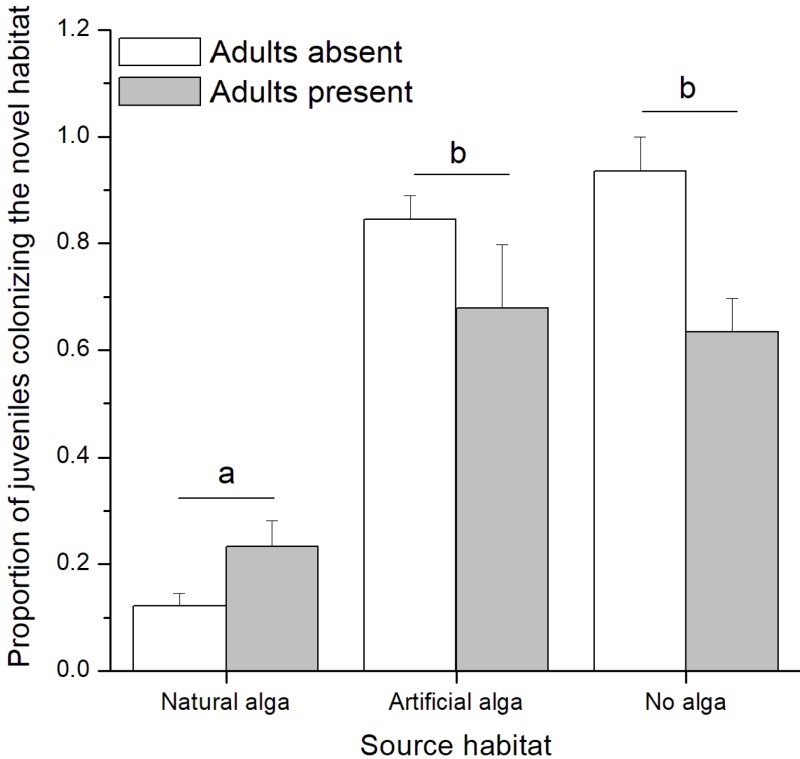

**Figure 4 Proportion of juveniles of *Cymadusa filosa* colonizing the novel habitat (natural alga with or without adults) when different source habitats (natural algal thallus, artificial algal thallus and none) were provided to the juveniles in the laboratory.** Bars represent standard error. Different letters indicate significant difference among juveniles' source habitats (Tukey test, *p* < 0.05).

## DISCUSSION

Habitat quality does not seem to affect colonization by juveniles of *Cymadusa filosa* on a local scale and under natural conditions (i.e., field experiment). However, on a smaller scale under controlled conditions (i.e., laboratory experiment), juveniles are able to optimize their habitat choice by selecting a high-quality resource (i.e., natural alga), where both food and shelter are provided, when low-quality resource (i.e., artificial alga) is available as a source or novel habitat. The colonization by juveniles did not depend on the algal patch size. The presence of conspecific adults does not affect the colonization of

juveniles under controlled conditions but may be important under natural conditions. These findings suggest that, although very initially constrained by their mothers' choice, *C. filosa* juveniles can rapidly search, choose and colonize novel habitats, corroborating their previously reported dispersive potential (*Bueno & Leite, 2019*), and that such process can be partially affected by the habitat quality.

In the laboratory, juveniles showed high colonization rates when going from artificial source habitats to natural novel habitats. In contrast, the lowest colonization rates were found when an artificial algal thallus was offered (i.e., novel habitat) for juveniles inhabiting a natural alga. In accordance, intermediate colonization rates were found when source and novel habitats were similar (i.e., both were natural or artificial algal thalli). These results suggest that *C. filosa* juveniles have a tendency to stay where they are, as expected since they are released from the mother brood pouch, unless a better-quality resource is available, corroborating the low mobility behavior often observed for tube-building amphipods (*Duffy & Hay, 1994*; *Poore & Steinberg, 1999*). Similar results were found in the laboratory experiment testing the effect of adult's presence. In this case, natural algal thalli (with or without adults) were offered as novel habitat in all treatments and juveniles with an artificial alga or without any alga (i.e., low-quality source habitats) showed higher colonization rates than those with a natural alga as source habitat. In this sense, our results corroborate the idea that the distribution of juvenile amphipods living on macroalgae is not restricted by the females' choice (*Poore, 2005*; *Bueno & Leite, 2019*) and that their colonization process is partially motivated by the habitat quality (*Poore, 2004*).

*Sargassum* is a high-quality food for *C. filosa* and, thus, is not only used for shelter by this herbivorous amphipod (*Machado, Siqueira & Leite, 2017*, *Machado, Ferreira & Leite, 2019*). This role for *Sargassum* is supported by our results from laboratory experiments with artificial and natural algal thalli. Also, these results suggest that *C. filosa* juveniles can detect a suitable host to colonize. Similarly, the tube-building amphipod *Jassa herdmani* can discriminate between its host hydrozoan *Tubularia indivisa* and other artificial substrates, indicating this species may have a detection mechanism, although it is not fully understood (*Havermans et al., 2007*). Moreover, the amphipod *Incisocalliope symbioticus* seems to use chemical cues to locate and colonize its host, the gorgonian octocoral *Melithaea flabellifera*, at short distances (*Kumagai, 2006*). Chemical cues from *Sargassum* may be an important mechanism for *C. filosa* juveniles detect a suitable and high-quality habitat and further studies are required to elucidate it.

The ability to detect and colonize a suitable habitat should be advantageous for *C. filosa* juveniles when (1) there is a mismatch between feeding (or habitat) preference of adults and performance of juveniles (*Cruz-Rivera & Hay, 2001*; *McDonald & Bingham, 2010*) and/or (2) the abundance of adults on high-quality food algae is constrained by extrinsic factors to the host, such as predation and wave action (*Sotka, 2007*; *Lasley-Rasher et al., 2011*; *Machado, Ferreira & Leite, 2019*). The feeding behavior and its consequences for the survival, growth and reproduction of *C. filosa* are tightly linked, with adults preferring algal hosts that results in the highest performance of juveniles (*Machado, Siqueira & Leite, 2017*, *Machado, Ferreira & Leite, 2019*). However, in the study area, *C. filosa* has been found on the red alga *Dichotomaria marginata*, which is a poor-quality food, in densities

higher than on the high-quality food *Sargassum filipendula* (*Machado, Ferreira & Leite, 2019*), suggesting a trade-off between food and shelter for this herbivorous amphipod. In this context, the ability of *C. filosa* juveniles to detect and colonize novel and suitable habitats, regardless their mothers' choice, should be advantageous for the success of this developmental stage and, consequently, for the species.

Contrasting with our expectations and the results from the laboratory experiments, we did not find any effect of habitat quality on juveniles' colonization in the field experiment, suggesting other factors may influence the colonization of these amphipods under natural conditions. The similar abundance of *C. filosa* juveniles on artificial and natural algal thalli may suggest that the algal morphology per se has an important role regarding the host use by this amphipod in the field. Algal morphology mediates the vulnerability of associated fauna to predation (*Zamzow et al., 2010*; *Ware et al., 2019*). Also, structural and spatial traits of macroalgae affects the availability of space for the epifauna (*Hacker & Steneck, 1990*; *Carvalho et al., 2018*). The colonization of artificial substrates mimicking natural plants by epifaunal species highlights the importance of the physical structure of macrophytes for the settlement of associated fauna (*Virnstein & Curran, 1986*; *Norderhaug, Christie & Rinde, 2002*; *Waage-Nielsen, Christie & Rinde, 2003*). Because artificial algal thalli in the field were rapidly colonized by juvenile amphipods, it is probably functioning as a suitable shelter, at least temporarily.

The size of algal thalli available for colonization in the field experiment did not affect the number of *C. filosa* juveniles found on these substrates. In a previous laboratory experiment, *C. filosa* juveniles colonized small and large algal patches in similar proportions (*Bueno & Leite, 2019*). The availability of substrate is often related to the habitat complexity, thus is an important factor for the association of mobile fauna with macrophytes (*Stoner, 1980*; *Hacker & Steneck, 1990*). For instance, as the surface area of a substrate increase, it can decrease the vulnerability of mobile fauna to predation (*Russo, 1987*). However, prey size may also mediate the intensity of predation on amphipods. Predators may preferentially consume larger individuals (*Sturaro et al., 2016*) while smaller animals may be less vulnerable to detection and consumption by predators (*Schlacher & Wooldridge, 1996*). In this sense, although habitat size is important for the associated fauna, the surface area may be less crucial for the colonization of a *Sargassum* thallus by small bodied organisms, such as *C. filosa* juveniles.

No effect of the presence of conspecific adults on the colonization of *C. filosa* juveniles was detected in the laboratory experiment. In contrast, we found a marginally significant trend of juveniles' abundance increasing with adult density. The effects of conspecifics on the recruitment and settlement of juvenile amphipods are variable. The presence of adults does not affect the habitat choice of juveniles of the amphipod *Pontogammarus robustoides* (*Czarnecka, Kobak & Wiśniewski, 2010*). For the tube-dwelling amphipod *Corophium volutator*, high densities of adults may negatively affect the settlement of conspecific juveniles, as a result of intraspecific competition (*Wilson, 1989*; *Jensen & Kristensen, 1990*). The effects of intraspecific competition depend on the density of conspecifics and food concentration (*Hill, 1992*; *Wenngren & Ólafsson, 2002*; *Van Tomme et al., 2012*). In our field experiment, we used treatments with adults in densities (from ~1 to 2 ind/g of

alga) that were much higher than that previously reported for *C. filosa* at same study site (~0.2 ind/g of alga; see *Machado, Ferreira & Leite, 2019*). Yet, the presence of adult conspecifics on algal patches did not negatively affect juveniles' colonization, suggesting that intraspecific competition for food and/or space is not an important factor for the colonization of *C. filosa* juveniles. In contrast, it is possible that *C. filosa* juveniles are attracted by the presence of adults, which could explain the increase of juveniles with adult density. *Corophium volutator* juveniles may be attracted by conspecific aggregations on intertidal soft-bottom habitats, which could be an indicative of habitat quality (*Bringloe et al., 2013*; *Drolet et al., 2013*). Further experiments with higher replication are necessary to confirm this hypothesis for *C. filosa* juveniles.

## CONCLUSIONS

By combining laboratory and field experiments, we found that the colonization of novel habitats by *C. filosa* juveniles can be affected by the habitat quality, but not by the patch size. Also, the presence of conspecifics may affect the colonization by juveniles. Contrasting results between laboratory and field experiments manipulating the habitat quality and the presence of conspecifics highlighted the importance of using different approaches to better understand the factors determining the host use by juvenile amphipods. In the present study, we used artificial algal thalli as a low-quality habitat; it would be interesting to investigate the behavior of *C. filosa* juveniles when they have more than one macroalgal host to select, since in the shallow subtidal areas they have a variety of biological substrates available for colonization (*Appadoo & Myers, 2004*; *Bueno, Dias & Leite, 2017*; *Machado, Ferreira & Leite, 2019*). Investigating dispersal patterns during specific life stages can greatly contribute to understand the movement paths during the lifespan of organisms (*Allen, Metaxas & Snelgrove, 2018*). After dispersing, the colonization of new substrates will set the distribution of populations among hosts (*Poore, 2005*; *Chapman, 2007*), contributing to the maintenance of populations at local and regional scales.

## ACKNOWLEDGEMENTS

We thank Ana P. Ferreira and Ricardo Ota for the assistance with field and laboratory work, and Edson A. Vieira for valuable comments.

### Funding

This study was financially supported by the Fundação de Amparo à Pesquisa do Estado de São Paulo as a post-doc fellowship granted to MB (2015/10797-9). The funders had no role in study design, data collection and analysis, decision to publish, or preparation of the manuscript.

### Grant Disclosures

The following grant information was disclosed by the authors:
Fundação de Amparo à Pesquisa do Estado de São Paulo: 2015/10797-9.

## Competing Interests

The authors declare that they have no competing interests.

## Author Contributions

- Marilia Bueno conceived and designed the experiments, performed the experiments, analyzed the data, prepared figures and/or tables, authored or reviewed drafts of the paper, and approved the final draft.
- Glauco B.O. Machado performed the experiments, analyzed the data, prepared figures and/or tables, authored or reviewed drafts of the paper, and approved the final draft.
- Fosca P.P. Leite conceived and designed the experiments, authored or reviewed drafts of the paper, and approved the final draft.

## Field Study Permissions

The following information was supplied relating to field study approvals (i.e., approving body and any reference numbers):

SISBIO/ICMBio approved the study (51999-1).

## Data Availability

Our raw data is available as a Supplemental File.

## Supplemental Information

Supplemental information for this article can be found online at http://dx.doi.org/10.7717/peerj.10188#supplemental-information.

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
