# Peer review of "Colonization of novel algal habitats by juveniles of a marine tube-dwelling amphipod"

_PeerJ, doi:10.7717/peerj.10188_

## Round 0.1 · original submission · Minor Revisions

I now have reviews from two specialists in the area of marine algal mesograzers, and while they are both positive about this submission, they also have a number of issues that they would like addressed before your contribution is published. Please consider each of the reviewers' comments in a point-by-point manner as you write your revised manuscript.

Reviewer 1 ·

Basic reporting

The article is well written in general but can still use the help of a language editor. It aslo needs to update and broaden the literature cite. There is too much self citation.

Experimental design

The hypotheses make sense when the paper is read completely, but the experimental questions need to be stated more clearly.
I have no issues with the experimental design or analysis other than some clarifications requested. I find the number of figures and their clarity appropriate.

Validity of the findings

Only one of the conclusions (effects of adults) requires further discussion. However, overall, the conclusion will be improved if they are placed in a broader context than just the amphipod or marine literature.

Additional comments

Review of Bueno et al. “Colonization of novel algal habitats…” PeerJ

General comments:
1) Overall, this well written, but there are some slight mistakes in phrasing and the use of articles. A language editor should give this a reading.
2) The authors are dealing with fundamental ecological issues such as the roles of dispersal and recruitment in population connectivity and community structure, age-related dispersal and other broader ecological questions. Most of the fundamental work on such questions has been done in terrestrial systems, especially with mammals and birds. It is surprising that so little of the relevant literature is cited. The manuscript will have broader appeal if the citations are less biased towards marine work in these fundamental issues.
3) On a related issue, the literature cited is not current. For example, there are more recent reviews on the ecology of dispersal, including supply-side processes, than the ones cited here. In fact, all papers from 2015 on are from one of the coauthors. I think there is more recent relevant literature than this.
4) Similarly, instances of repetitive citing should be avoided (see for example Mungia et al. 2007 in lines 97-99).

Specific comments:
Lines 46-48: the methods are just a repetition of the background, please rephrase one or the other. Also “settlement” invokes a specific type of movement. Please use colonization or recruitment as these amphipods are likely crawling, rather than swimming.
Line 113: Eliminate “known as”
Lines 131-145: The presentation of the hypotheses or experimental questions must be written more clearly. The authors use multiple vague statements and terms here that are not helpful (e.g., “the macroalgae-amphipod system”, “this early stage”). They also use several references when establishing their actual hypotheses, which is confusing. Are these hypotheses tested in those papers? The hypotheses and their justifications should be stated more directly.
Line 159: Deleted “the amphipod” and abbreviate the genus.
Line 163: “thalli” should not be in italics
Line 166: “Driven”, not “drove”
Lines 165-170: The experimental units are difficult to visualize so I recommend adding a picture or diagram as supplementary material. Also, it is unclear what the components of the unit are for. Please explain.
Line 172: Eliminate “of experiment”
Line 180-181: This sentence is unclear. What epibionts?
Line 184-194: Please eliminate superfluous details and explain more clearly. I am not sure what a replicate means from this
Lines 319-320: I am not sure what this means. All amphipods are capable of colonizing substrates pending on circumstances.
Lines 325-326: I am not convinced the presence of adults was irrelevant. The result of their analysis was borderline (p=0.055) and their replication was relatively low. The authors should comment on this.
Lines 366-368: I understand the trade-off but the role of juvenile mobility should be explained better in this context.

Reviewer 2 ·

Basic reporting

No comment

Experimental design

No comment

Validity of the findings

1- I have one comment on the statistics of the field experiment with the presence/absence of adults and its interpretation. There was a marginally-significant influence of adult male density and colonization rate (p=0.055); there was a 2-fold increase in juvenile recruits in cups with 8 adults relative to 0 adults (lines 306-309). The authors interpret this as no effect, but to me it is also possible, even likely, the experiment suffers from Type II error and low power. Id like the authors to mention this in either the results or Discussion. At the least, the Discussion should point out that the effect was marginally significant (e.g., paragraph starting line 392) - in fact, id suggest placing these results into a new figure. they are hard-fought data and deserve more attention.

I'd also recommend the authors try a couple of analyses to pull out this effect. Does the analysis treat the male density as ordinal levels, or does it ignore the increasing levels of male density? Also, you might want to treat the levels as presence / absence (i.e., group 4 and 8 males), since that likely increases your power.

2- a few issues with the artificial alga came to my head. Were artificial algae pre-treated in the water before any colonization experiments? If they were colonized by epibiota that can attract herbivorous amphipods (e.g., diatoms), then that would suggest they have both food and habitat quality. Also, microbial biofilms can generally influence invertebrate habitat choices. Finally, plastic can leach chemicals that can either repel or harm marine organisms. What precautions did you take to prevent this?

3- there were substantial numbers of non-Cymadusa filosa juveniles in both field experiments. Were there any patterns between treatments in these other groups, or as a whole? If you know if any of these are competitors or predators, Id be particularly curious to know on this point, as it could help to explain the lack of field results.

Additional comments

This is a well-executed set of laboratory and field experiments exploring the ecology of juvenile colonization in a species of herbivorous amphipod. Overall, the experiments showed that these juveniles make explicit choice to colonize algal habitats over simple morphological structure, and that the effect of adult male presence on colonization rates was weaker. there was a marginally-significant increase in juvenile colonization in the field in the presence of adult males.

---

## Round 0.2 · accepted · Accept

The authors have addressed the concerns of both reviewers and considered the statistical issues raised by one of them with subsequent revisions of the ms.